# Seroprevalence of Specific Antibodies to *Toxoplasma gondii*, *Neospora caninum*, and *Brucella* spp. in Sheep and Goats in Egypt

**DOI:** 10.3390/ani12233327

**Published:** 2022-11-28

**Authors:** Ragab M. Fereig, Gamal Wareth, Hanan H. Abdelbaky, Amira M. Mazeed, Mohamed El-Diasty, Adel Abdelkhalek, Hassan Y. A. H. Mahmoud, Alsagher O. Ali, Abdelrahman El-tayeb, Abdullah F. Alsayeqh, Caroline F. Frey

**Affiliations:** 1Division of Internal Medicine, Department of Animal Medicine, Faculty of Veterinary Medicine, South Valley University, Qena 83523, Egypt; 2Friedrich-Loeffler Institut, Institute of Bacterial Infections and Zoonosis, Naumburger Str. 96a, 07743 Jena, Germany; 3Doctor of Veterinary Sciences, Veterinary Clinic, Veterinary Directorate, Qena 83523, Egypt; 4Department of Infectious Disease, Faculty of Veterinary Medicine, Arish University, Arish 45516, Egypt; 5Agricultural Research Center (ARC), Animal Health Research Institute-Mansoura Provincial Lab (AHRI-Mansoura), P.O. Box 264-Giza, Cairo 12618, Egypt; 6School of Veterinary Medicine, Badr University, Cairo 11829, Egypt; 7Division of Infectious Diseases, Department of Animal Medicine, Faculty of Veterinary Medicine, South Valley University, Qena 83523, Egypt; 8Educational Veterinary Hospital, Faculty of Veterinary Medicine, South Valley University, Qena 83523, Egypt; 9Department of Veterinary Medicine, College of Agriculture and Veterinary Medicine, Qassim University, Buraidah 51452, Saudi Arabia; 10Institute of Parasitology, Department of Infectious Diseases and Pathobiology, Vetsuisse-Faculty, University of Bern, Länggassstrasse 122, CH-3012 Bern, Switzerland

**Keywords:** toxoplasmosis, neosporosis, brucellosis, ovine, caprine, Egypt

## Abstract

**Simple Summary:**

*Toxoplasma gondii*, *Neospora caninum* and *Brucella* species are widely distributed infectious agents that induce severe clinical illness in numerous farm animals. In addition *T. gondii* and *Brucella* spp. are of public health concern because of their zoonotic potentials. In Egypt and other countries, toxoplasmosis, neosporosis, and brucellosis are three well-known etiological agents of infectious abortion in many farm animals including sheep and goats. Inaccurate diagnosis of the abortion cause results in great economic losses because of application of inappropriate control measures. In the current study, we provided the first report on the comparative seroprevalence of toxoplasmosis, neosporosis and brucellosis in sheep and goats in Egypt based on antibody detection. We found that the overall seroprevalence rate was the highest for *T. gondii*. Nevertheless, only antibodies to *Brucella* spp. were associated with recent abortion in sheep and goats. Our study provides solid data on the existence of abortifacient agents in the common small ruminants in Egypt. Accordingly, these data can be exploited in developing more efficient control policies against such serious infections.

**Abstract:**

Toxoplasmosis, neosporosis, and brucellosis are devastating diseases causing infectious abortion and, therefore, substantial economic losses in farm animals. Toxoplasmosis and neosporosis are caused by the intracellular protozoan parasites *Toxoplasma gondii* (*T. gondii*) and *Neospora caninum* (*N. caninum*), respectively. Brucellosis is a bacterial disease caused by numerous *Brucella* species in multiple hosts. Toxoplasmosis and brucellosis are also considered foodborne zoonotic diseases. In the current study, specific antibodies to *T. gondii* and *N. caninum,* in addition to those to *Brucella* spp., were detected to gain a better understanding of the epidemiological situation for these three pathogens. Sheep and goat sera from Egypt (*n* = 360) of animals with and without a history of abortion were tested using commercial ELISAs. Seropositivity rates of 46.1%, 11.9%, and 8.6% for *T. gondii*, *N. caninum*, and *Brucella* spp., respectively, were revealed. Mixed infections with *T. gondii* and *Brucella* spp. (4.4%), *T. gondii* and *N. caninum* (4.2%), *N. caninum* and *Brucella* spp. (1.4%), and even some triple infections (0.6%) have been observed. Animals with a history of abortion had a significantly higher seroprevalence for *Brucella* spp. infection than those without abortion (12.6%; 28/222 vs. 2.2%; 3/138) (*p* = 0.0005; Odds ratio = 1.9–21.8), while none of the other pathogens showed a similar effect. This result suggests brucellosis as a possible cause of abortion in the study population. However, the high seroprevalence for *T. gondii* and *N. caninum* revealed in our study warrants further investigations.

## 1. Introduction

*Toxoplasma gondii* (*T. gondii*) is an intracellular protozoan parasite that affects almost all warm-blooded animals, including cats as definitive hosts as well as sheep, goats, pigs, and humans as intermediate hosts. The vertical transmission from dams to offspring, ingestion of food or water contaminated with oocysts excreted by infected cats, or ingestion of uncooked or undercooked meat containing tissue cysts are the major modes of transmission of *T. gondii* [1]. 

*Neospora caninum* (*N. caninum*) is a closely related protozoan parasite to *T. gondii* and affects dogs as final and intermediate hosts and numerous animal species, particularly cattle and sheep as intermediate ones. Neosporosis is characterized clinically by neurological disorders and reproductive problems in dogs and mainly abortion in other intermediate hosts. Transmission is remarkably similar to *T. gondii* and occurs either by horizontal ingestion of oocysts or tissue cysts or vertically from infected dams to their offspring [1,2]. Up-to-date, clinical neosporosis in humans has not been confirmed yet, despite the detection of specific antibodies in serum samples of human origin [3]. However, a recent study found *N. caninum* DNA in human umbilical cord blood samples but not in the respective placentas [4]. 

*Brucella* (*B*.) spp., Gram-negative coccobacillus bacteria, are also common aborting agents in multiple animal species, especially sheep, goats, cattle, buffaloes, and pigs. *Brucella* includes many species, of which *B. abortus*, *B. melitensis*, and *B. suis* are the most common among farm animals. *Brucella melitensis* is the most important one in sheep and goats, and abortion mainly occurs in case of infection of pregnant ewes or goats [5]. Transmission of *Brucella* organisms usually occurs through direct or indirect contact with infected animals or infected materials or ingestion of contaminated feed or water [6].

Although *T. gondii* and *N. caninum* are protozoan parasites and *Brucella* spp. are bacteria, they possess several similarities in aspects of infection and pathogenesis. All these pathogens are intracellular microorganisms that evade the killing by macrophages and even use them to reach their target tissues [7,8,9,10,11].

A placental and/or fetal infection caused by *T. gondii*, *N. caninum*, or *B*. *melitensis* in pregnant sheep and goats may result in fetal death, abortion, or stillbirth, as well as severe placentitis in the case of brucellosis [12,13,14,15,16,17]. In addition, *Brucella* spp. cause a decrease in progesterone production and an increase in estrogen levels, resulting in premature delivery [18]. Abortion induced by the three above-mentioned infectious agents is a major cause of substantial economic losses in livestock production. Financial losses have been estimated for abortions in sheep caused by *T. gondii* to amount to over EUR 100/abortion in Spain [19], and for abortions in sheep and goats caused by *Brucella* sp. to amount to over USD 120 million annually in India [20]. Brucellosis and toxoplasmosis not only cause enormous economic losses, but they also are of substantial public health concern [1,21,22]. Brucellosis is mainly a serious occupational hazard (characterized by variable clinical signs ranging from fever to miscarriage [23,24,25]) for people in contact with infected animals or contaminated animal products, byproducts, or utensils, such as farm workers, animal owners or dealers, veterinarians, and abattoir or laboratory workers. 

Numerous animal species, including sheep and goats, are susceptible to toxoplasmosis, neosporosis, and brucellosis in Egypt [26,27,28,29,30]. However, simultaneous investigation of antibodies to *T. gondii*, *N. caninum*, and *Brucella* spp. was not conducted before in Egypt, particularly in sheep and goats. Thus, this study aimed to estimate the prevalence of these infections in sheep and goats from various regions of Egypt and assess correlations with the abortion history of the sampled animals. 

## 2. Materials and Methods

### 2.1. Ethical Statement

This study was conducted according to instructions established by the “Research Board” of the Faculty of Veterinary Medicine, South Valley University, Qena, Egypt. The protocols were approved by the Research Code of Ethics at South Valley University number 36 (RCOE-36). Blood samples were collected by a group of highly trained veterinarians and staff after consultation with the officials and animal owners.

### 2.2. Description of the Animals and Regions of the Study 

The present study was conducted in four governorates representing the northern part (Dakahlia), middle part (Beni Suef), southern part (Qena), and eastern part (Red Sea) of Egypt (Figure 1). 

The study animals were indigenous sheep and goats kept in farms (more than fifty animals) or small holdings (<50 animals/farm). Samples were collected at various time points from May 2016 to November 2021 (Table 1). All sheep and goats included in this study were older than 6 months and mostly females. None of the sampled animals was vaccinated against *Brucella* infection, as confirmed by animal owners. A total of 360 serum samples were collected randomly from sheep (*n* = 239) and goats (*n* = 121) with or without a history of abortion. Sera collected from animals with a history of abortion were confined mostly to medium to large farms (50 to 200 heads/farm) and collected during the first few months after abortions occurred. The selection of samples for the study was designed to cover a large time span (5 years) and a wide geographical range (approximately 1000 kilometers apart) in order to give a reliable estimation of the epidemiological situation of the three pathogens. Furthermore, flocks with and without a history of abortion were included, while vaccination for *Brucella* was an exclusion criterium. Within the selected flocks, samples were collected randomly from apparently healthy animals. 

### 2.3. Sample Collection and Serum Preparation

Approximately 5 mL of whole blood samples were collected by venipuncture from the jugular vein using sterile disposable plain vacutainer tubes and needles. The blood samples were kept in an upward position and allowed to clot at room temperature for 2–3 h. The samples were transported on ice in an icebox to our laboratory at the Faculty of Veterinary Medicine, South Valley University, Egypt. Sera were transferred into 1.5 mL Eppendorf tubes after centrifugation at 10,000× *g* for 15 min [30] and stored at −20 °C until use.

### 2.4. Serological Diagnosis of T. gondii, N. caninum, and Brucella Species

In order to detect antibodies to *T. gondii*, the serum samples were analyzed with the indirect multi-species ELISA for toxoplasmosis (ID.vet, Grabels, France) according to the manufacturer’s instructions. Serum samples and controls were diluted 1:10. The optical density (OD) obtained was used to calculate the percentage of sample (*S*) to positive (*P*) ratio (S/P%) for each of the test samples according to the following formula: S/P (%) = (OD sample − OD negative control)/(OD positive control − OD negative control) × 100. Samples with an S/P% less than 40% were considered negative; if the S/P% was between 40% and 50%, the result was considered doubtful and considered positive if the S/P% was greater than 50%.

For antibodies to *N. caninum*, serum samples were analyzed with the competitive multi-species ELISA for neosporosis (ID.vet, Grabels, France). Serum samples and controls were diluted 1:2. The ODs obtained were used to calculate the percentage of sample (*S*) to negative (*N*) ratio (S/N%) for each of the test samples according to the following formula: S/N (%) = OD sample/OD negative control × 100. Samples with an S/N% greater than 60% were considered negative; if the S/N% was between 50% and 60%, the result was considered doubtful, and it was considered positive if the S/N% was less than 50%. 

Concerning *Brucella* spp., serum samples were analyzed with the indirect multi-species ELISA for brucellosis (ID.vet, Grabels, France). Serum samples and controls were diluted 1:20. The ODs obtained were used to calculate the percentage of sample (*S*) to positive (*P*) ratio (S/P %) for each of the test samples according to the following formula: S/P (%) = (OD sample − OD negative control)/(OD positive control − OD negative control) × 100. Samples with an S/P% less than 110% were considered negative; if the S/P% was between 110% and 120%, the result was considered doubtful, and it was considered positive if the S/P% was greater than or equal to 120%. 

The ODs of all ELISA results were read at 450 nm and measured with an Infinite^®^ F50/Robotic ELISA reader (Tecan Group Ltd., Männedorf, Switzerland). More details for the commercial kits used are available in Table 2.

### 2.5. Statistical Analysis

The significance of the differences in the prevalence rates and risk factors was analyzed with Chi-square (Pearson) test, 95% confidence intervals (including continuity correction), and odds ratios using an online statistical website www.vassarstats.net (accessed on 9 August 2022). *p*-values and odds ratio were also confirmed with GraphPad Prism version 5 (GraphPad Software Inc., La Jolla, CA, USA). The results were considered significant when the *p*-value was <0.05 or highly significant when the *p*-value was <0.0001.

## 3. Results

The highest overall seroprevalence was found for *T. gondii* (46.1%; 166/360), subdivided as 35.6% (85/239) in sheep and 66.9% (81/121) in goats (Table 3). For *N. caninum*, a seroprevalence of 11.9%, 15.5%, and 5% in all animals, sheep, and goats, respectively, was found. Antibodies to *Brucella* spp. were detected in 8.6%, 10.5%, and 5% of all animals, sheep, and goats, respectively. Regarding mixed infection, seropositive rates in the three categories (all, sheep, and goats) were reported for *T. gondii* and *N. caninum* (4.2%, 4.2%, and 4.1%), and *T. gondii* and *Brucella* spp. (4.4%, 4.6%, and 4.1%), respectively. Mixed *N. caninum* and *Brucella* spp. infections were only recorded in 2.1% (5/239) of the sheep, leading to an overall rate of 1.4% (5/360). In addition, two sheep from northern Egypt (Dakhalia) were seropositive for all three infectious agents (*T. gondii*, *N. caninum*, and *Brucella* spp.), resulting in a seroprevalence of 0.8% (2/239) in sheep and 0.6% (2/360) for all tested animals (Table 3).

Possible associations between antibodies to *T. gondii*, *N. caninum* and *Brucella* spp. and animal species (sheep and goat), location (Dakahlia, Beni Suef, Qena, and Red Sea governorates), and abortion history (yes or no) were investigated. For *T. gondii*, goats were more likely to be seropositive, with a prevalence of 66.9% in goats compared to 35.6% in sheep (odds ratio (OR) 3.7, *p* = < 0.0001). Moreover, the location was regarded as a predisposing factor for *T. gondii* infection with a higher prevalence in Qena (66.1%; OR = 15.9; *p* ≤ 0.0001), Beni Suef (43.9%; OR = 6.4; *p* = 0.0004), and Dakahlia (32.4%; OR = 3.9; *p* = 0.0051) compared to the seroprevalence in Red Sea (10.9%) set as a reference. History of abortion was not associated with antibodies against *T. gondii* (*p* = 0.2435) (Table 4). 

Regarding risk factor analysis for *N. caninum*, the seroprevalence in sheep (13.1%) was significantly higher than that in goats (5%) (OR = 0.3; *p* = 0.0036). Regarding the location, the prevalence in Dakahlia (16.2%; OR = 2.7; *p* = 0.0130) and Red Sea (23.9%; OR = 4.3; *p* = 0.0009), but not Beni Suef (7.3%; OR = 1.4; *p* = 0.6391), were significantly higher than that of Qena (6.8%). The history of abortion was not associated with antibodies to *N. caninum* (*p* = 0.8624) (Table 5).

For *Brucella* risk factor assessment, the seroprevalence in sheep (10.5%) was higher than that in goats (5%), although this was not statistically significant (OR = 0.5; *p* = 0.0787). Samples collected from Dakahlia had a significantly higher seroprevalence (18.9%; OR = 4.5; *p* = 0.0002) than those from Beni Suef (4.9%; OR = 1; *p* = 0.9873), Red Sea (0%; OR = 0.2; *p* = 0.1243), or from Qena (4.9%). Other than for *T. gondii* and *N. caninum*, the history of recent abortion was correlated with antibodies against *Brucella* spp. Animals with a history of recent abortion had a significantly higher seroprevalence (12.6%; OR = 6.5; *p* = 0.0005) than sheep and goats without a history of abortion (2.2%) (Table 6). 

## 4. Discussion

*Toxoplasma gondii*, *N. caninum*, and *Brucella* spp. are professional abortifacient infectious agents in several animal species and different countries. Studies on the seroprevalence of all three pathogens and possible association with abortion history in sheep and goats have not yet been conducted in Egypt. Brucellosis is quite a common occupational disease among veterinarians and animal handlers in Egypt, and it is usually regarded as the first cause of infectious abortion in ruminants, especially in late gestation, and complicated by retention of the placenta [23]. These assumptions are supported by studies and official investigations, which diagnosed brucellosis as a cause of abortion in many abortion outbreaks [26,31,32]. However, our study of comparative seroprevalence of *T. gondii*, *N. caninum* and *Brucella* species in small ruminants had two main drivers: First, most of the reported brucellosis investigations were performed using screening tests, such as the Rose Bengal test and serum agglutination test, not confirmatory tests such as complement fixation test or ELISA. Second, numerous previous studies have recorded a high prevalence of *T. gondii* and *N. caninum* antibodies in many farm animals in Egypt, including in small ruminants, and therefore their contribution to abortions might currently be underestimated [27,28,30]. 

Our study indeed found the highest seroprevalence against *T. gondii* among all three tested pathogens, thus confirming previous studies from Egypt that found seroprevalences between 28.7% and 98.4% in small ruminants [33,34,35]. In our study, goats were significantly more affected (66.9%) than sheep (35.6%), and a region with a lower seroprevalence was also identified (Red Sea, eastern Egypt, compared to northern, middle, and southern Egypt). These findings confirmed earlier results from Egypt in that region and animal species have an effect on *T. gondii* seroprevalence [35], and also other studies found similar effects, namely Magalhães et al. [36] and Rêgo et al. [37], in Brazil; Dong et al. in China [38]; and Suazo-Cortez et al. in Mexico [39]. As for *N. caninum*, the seroprevalence in our study was 11.9% overall, 15.5% in sheep, and 5% in goats, respectively. Our results compared nicely with previous reports from sheep in Egypt: 8.6% reported by Selim et al. [30], and 36.1% reported by El-Ghaysh et al. [40]. Moreover, for this parasite, animal species and location had significant effects on the seroprevalence, with sheep being more affected than goats and northern and eastern Egypt being more affected than middle and southern Egypt. Other authors have also reported significant effects of location and animal species on *N. caninum* seroprevalence in small ruminants [30,41,42,43]. 

Of all three pathogens tested, seroprevalence against *Brucella* spp. was the lowest, with 8.6% overall, 10.5% in sheep, and 5% in goats, respectively. Other authors found prevalences varying between 7.2% and 56.7% for sheep and goats in Egypt [29,32,44,45]. In our study, there was a trend of higher seropositivity in sheep compared to goats, but it was not significant. Northern Egypt, however, did have a significantly higher seroprevalence than the other regions tested. Similar factors were found before by other authors from Egypt [46,47,48] and elsewhere [49,50,51]. 

Arguably, the most interesting association revealed in this study was the seropositivity against *Brucella* spp. and a history of abortion. Interestingly, no such association was found between antibodies against *T. gondii* or *N. caninum* and recent abortions. A study conducted on pregnant ewes in Iraq found that ewes seroconverting to *Brucella* were 2.9 times more likely to lose their pregnancy than those remaining seronegative. In the same study, seroconverting to *T. gondii* had no significant impact on the loss of pregnancy [52]. Similarly, in another region of the world, only about 1.6% of lost lambs were attributed to *T. gondii* in a prospective cohort study in primiparous ewes from southern Australia, albeit at an overall much lower seroprevalence than in our study [53]. Moreover, in a study performed with breeding ewes from Mexico, seroprevalence for *T. gondii* (61.96%) and *N. caninum* (15.22%) was similar to our study, and *T. gondii* seroprevalence was not associated with a history of abortion, but *N. caninum* seroprevalence was [54]. However, although our results suggest an involvement of *Brucella* in the abortions, they do not negate the role of *T. gondii* or *N. caninum* as aborting agents. Numerous previous reports have documented toxoplasmosis [1,19,55] and neosporosis [56,57,58] as causes of abortion in sheep and goats. 

The high prevalence of anti-*Brucella* antibodies in animals with an abortion history was correlated to the highest seropositivity in Dakahlia governorate in the Nile Delta region, northern Egypt. This area is endemic for different *Brucella* species, including *B. abortus*, the causative agent of bovine brucellosis, and *B. melitensis*, the main cause of ovine and caprine brucellosis [26,29]. In addition, the case history of abortion in that region demonstrated that some aborting ewes and does have experienced late abortion and retained placenta, both common signs of *Brucella* abortion [12,23]. In contrast, *Neospora* abortion can occur at any time with a special preference for mid-gestation, and usually, no clinical signs or lesions are evident in aborting dams or in the placenta or fetus [16]. In the case of toxoplasmosis, early abortion and fetal resorption are mostly noticed, and similarly to neosporosis, no abnormalities in the dams, fetus, or placenta are usually observed [21].

However, to obtain a better understanding of the causes of abortions of small ruminants in Egypt, fetuses, placentas, and serum samples of the mothers should be submitted for laboratory analyses. Moreover, our data provide insights into the current epidemiological situation for *Brucella* spp. and *T. gondii*, both important zoonotic species, and *N. caninum*, all three of which cause losses to the livestock industry. More efficient control programs will have to be established in the future based on a good understanding of the real situation, which can only be gained by investigations such as this one.

Due to the limitations in the number, content, and distribution of studies focusing on the relevant theme worldwide, including Egypt, we could not perform a comprehensive comparison with our study. Additionally, higher numbers of samples are required from sheep, goats, and other species and geographic locations to establish an accurate assessment of the main causes of abortions among our tested pathogens.

## 5. Conclusions

Abortion in farm animals, including sheep and goats, is a major and direct cause of enormous economic losses in animal production sectors. Toxoplasmosis, neosporosis, and brucellosis are well-known causes of infectious abortions in sheep and goats in Egypt and other countries. Misevaluation of the current situation of these infections as causes of abortion would jeopardize the livestock industry in Egypt. This study is the first to establish a comparison between toxoplasmosis, neosporosis, and brucellosis based on seroprevalence in sheep and goats in Egypt. Seroprevalence for *T. gondii* was highest, followed by *N. caninum* and *Brucella* spp. Animal species and location were considered as risk factors for *T. gondii* and *N. caninum* infections, while only the location was a significant factor for *Brucella* spp. infection. Importantly, antibodies to *Brucella* spp. were significantly higher in aborting than in non-aborting animals, a feature not observed for the other two pathogens.

## Figures and Tables

**Figure 1 animals-12-03327-f001:**
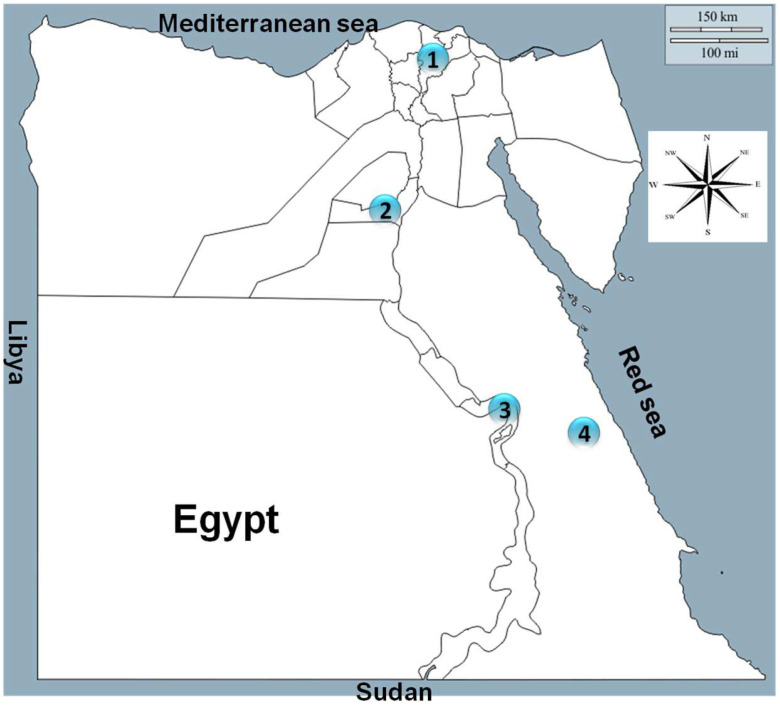
Geographic locations of the tested regions: 1—Dakahlia governorate (northern Egypt); 2—Beni Suef governorate (middle Egypt); 3—Qena governorate (southern Egypt); Red Sea governorate (eastern Egypt).

**Table 1 animals-12-03327-t001:** Sample information and history of abortion.

Sample Group	Time of Collection	Species	Number	Place of Collection	Type of Breeding System	History of Abortion
Group 1	2016	Sheep	111	Dakahlia	Farm	Yes
Group 2	2016	Sheep Goat	12 29	Beni Suef	Farm	Yes
Group 3	2021	Sheep	70	Qena	Farm	Yes
Group 4	2018	Goat	92	Qena	Smallholders	No
Group 5	2016	Sheep	46	Red Sea	Smallholders	No
Total	2016–2021	Sheep and goat	360	Various	Farm and smallholder	Various

**Table 2 animals-12-03327-t002:** Commercially available ELISA test kits used for detecting antibodies to *T. gondii*, *N. caninum*, and *Brucella* spp.

Infectious Agent	ELISA Test Kit	Manufacturer	Antigen	Conjugate	Sensitivity *	Specificity *
*Toxoplasma gondii*	ID Screen^®^ Toxoplasmosis Indirect Multi-species	ID.vet Innovative Diagnostics, Grabels, France	P30 antigen	Anti-multi-species IgG-HRP	98.36% (CI 95% ^#^: 95.29–99.44%)	99.42% (CI 95%: 98.8–100%)
*Neospora caninum*	ID Screen^®^ *Neospora caninum* competition Multispecies	ID.vet Innovative Diagnostics, Grabels, France	Purified extract of *Neospora caninum*	Anti-*N. caninum*-HRP (detect IgG or IgM)	100% (CI 95%: 98.8–100%)	100% (CI 95%: 99.63–100%)
*Brucella* species	ID Screen^®^ Brucellosis Serum Indirect Multispecies	ID.vet Innovative Diagnostics, Grabels, France	LPS of *Brucella* species	Anti-multi-species-IgG-HRP	100% (CI 95%: 89.57–100%)	99.74% (CI 95%: 99.24–99.91%)

* The sensitivity and specificity of the diagnostic kits were provided by the manufacturer of the kits. ^#^ CI 95%; 95% confidence interval.

**Table 3 animals-12-03327-t003:** Seroprevalence of *T. gondii*, *N. caninum*, *Brucella* spp., and mixed infections in sheep and goats from Egypt.

Type of Infection	Animal Species	No. of Tested	No. of Negative (%)	No. of Doubtful (%)	No. of Positive (%)	95% CI *
*T. gondii*	Sheep Goat Both	239 121 360	142 (59.4) 37 (30.6) 179 (49.7)	12 (5) 3 (2.5) 15 (4.2)	85 (35.6) 81 (66.9) 166 (46.1)	29.6–42 57.7–75.1 40.9–51.4
*N. caninum*	Sheep Goat Both	239 121 360	191 (79.9) 113 (93.4) 304 (84.4)	11 (4.6) 2 (1.6) 13 (3.6)	37 (15.5) 6 (5) 43 (11.9)	11.3–20.8 2–10.9 8.9–15.9
*Brucella* species *^#^*	Sheep Goat Both	239 121 360	206 (86.2) 112 (92.6) 318 (88.3)	8 (3.3) 3 (2.5) 11 (3.1)	25 (10.5) 6 (5) 31 (8.6)	7–15.2 2–10.9 6–12.1
*T. gondii + N. caninum*	Sheep Goat Both	239 121 360	219 (91.6) 114 (94.2) 333 (92.5)	10 (4.2) 2 (1.7) 12 (3.3)	10 (4.2) 5 (4.1) 15 (4.2)	2.1–7.8 1.5–9.8 2.4–6.9
*T. gondii + Brucella* species	Sheep Goat Both	239 121 360	227 (95) 114 (94.2) 341 (94.7)	1 (0.4) 2 (1.7) 3 (0.8)	11 (4.6) 5 (4.1) 16 (4.4)	2.4–8.3 1.5–9.8 2.6–7.3
*N. caninum + Brucella* species	Sheep Goat Both	239 121 360	230 (96.2) 120 (99.2) 350 (97.2)	4 (1.7) 1 (0.8) 5 (1.4)	5 (2.1) 0 5 (1.4)	0.8–5.1 0–3.8 0.5–3.4
*T. gondii + N. caninum + Brucella* species	Sheep Goat Both	239 121 360	234 (97.9) 120 (99.2) 354 (98.3)	3 (1.3) 1 (0.8) 4 (1.1)	2 (0.8) 0 2 (0.6)	0.2–3.3 0–3.8 0.1–2.2

* 95% CI calculated according to the method described by (http://vassarstats.net/ accessed on 9 August 2022); *^#^* the kit can detect antibodies against *Brucella abortus*, *B. melitensis*, and *B. suis*.

**Table 4 animals-12-03327-t004:** Risk factors for *T. gondii* antibodies in sheep and goats in Egypt.

Analyzed Factor	No. of Tested	No. of Negative (%)	No. of Positive (%)	OR (95% CI) *	*p*-Value ^#^
Animal species					
Sheep	239	154 (64.4)	85 (35.6)	Ref.	Ref.
Goat	121	40 (33.1)	81 (66.9)	3.7 (2.3–5.8)	<0.0001
Geographical location					
Dakahlia (Northern Egypt)	111	75 (67.6)	36 (32.4)	3.9 (1.4–10.8)	0.0051
Beni Suef (Middle Egypt)	41	23 (56.1)	18 (43.9)	6.4 (2.1–19.6)	0.0004
Qena (Southern Egypt)	162	55 (33.9)	107 (66.1)	15.9 (6–42.7)	<0.0001
Red Sea (Eastern Egypt)	46	41 (89.1)	5 (10.9)	Ref.	Ref.
Abortion history					
Yes	222	125 (56.3)	97 (43.7)	0.8 (0.5–1.2)	0.2435
No	138	69 (50)	69 (50)	Ref.	Ref.

* Odds ratio at 95% confidence interval as calculated by http://vassarstats.net/ (accessed on 9 August 2022); ^#^ *p*-value was evaluated by Chi-square test (Pearson test) using online statistics software http://vassarstats.net/ (accessed on 9 August 2022) and GraphPad Prism version 5; Ref.; is the value that used as a reference.

**Table 5 animals-12-03327-t005:** Risk factors for *N. caninum* antibodies in sheep and goats in Egypt.

Analyzed Factor	No. of Tested	No. of Negative (%)	No. of Positive (%)	OR (95% CI) *	*p*-Value ^#^
Animal species					
Sheep	239	202 (86.9)	37 (13.1)	Ref.	Ref.
Goat	121	115 (95)	6 (5)	0.3 (0.1–0.7)	0.0036
Geographical location					
Dakahlia (Northern Egypt)	111	93 (83.8)	18 (16.2)	2.7 (1.2–5.9)	0.0130
Beni Suef (Middle Egypt)	41	38 (92.7)	3 (7.3)	1.4 (0.4–5.2)	0.6391
Qena (Southern Egypt)	162	151 (93.2)	11 (6.8)	Ref.	Ref.
Red Sea (Eastern Egypt)	46	35 (76.1)	11 (23.9)	4.3 (1.7–10.8)	0.0009
Abortion history					
Yes	222	196 (88.3)	26 (11.7)	0.9 (0.5–1.8)	0.8624
No	138	121 (87.7)	17 (12.3)	Ref.	Ref.

* Odds ratio at 95% confidence interval as calculated by http://vassarstats.net/ (accessed on 9 August 2022); ^#^ *p*-value was evaluated by Chi-square test (Pearson test) using online statistics software http://vassarstats.net/ (accessed on 9 August 2022) and GraphPad Prism version 5; Ref.; value used as a reference.

**Table 6 animals-12-03327-t006:** Risk factors for *Brucella* spp. antibodies in sheep and goats in Egypt.

Analyzed Factor	No. of Tested	No. of Negative (%)	No. of Positive (%)	OR (95% CI) *	*p*-Value ^#^
Animal species					
Sheep	239	214 (89.5)	25 (10.5)	Ref.	Ref.
Goat	121	115 (95)	6 (5)	0.5 (0.2–1.1)	0.0787
Geographical location					
Dakahlia (Northern Egypt)	111	90 (81.1)	21 (18.9)	4.5 (1.9–10.6)	0.0002
Beni Suef (Middle Egypt)	41	39 (95.1)	2 (4.9)	1 (0.2–4.8)	0.9873
Qena (Southern Egypt)	162	154 (95.1)	8 (4.9)	Ref.	Ref.
Red Sea (Eastern Egypt)	46	46	0	0.2 (0–3.5)	0.1243
Abortion history					
Yes	222	194 (87.4)	28 (12.6)	6.5 (1.9–21.8)	0.0005
No	138	135 (97.8)	3 (2.2)	Ref.	Ref.

* Odds ratio at 95% confidence interval as calculated by http://vassarstats.net/ (accessed on 9 August 2022); ^#^ *p*-value was evaluated by Chi-square test (Pearson test) using online statistics software http://vassarstats.net/ (accessed on 9 August 2022) and GraphPad Prism version 5; Ref.; value used as a reference.

## Data Availability

All data generated and analyzed during this study are included in this published article. Raw data supporting the findings of this study are available from the corresponding author on request.

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
