# Peer review of "Seroprevalence of Specific Antibodies to *Toxoplasma gondii*, *Neospora caninum*, and *Brucella* spp. in Sheep and Goats in Egypt"

_animals, 2022, doi:10.3390/ani12233327_

Round 1

Reviewer 1 Report

Line No. 75-77 need to rephrase

Double-check the reference given in line No. 79 (Reference No. 18)

Lines 86-92 are not relevant to the topic. Should be deleted (In humans, toxoplasmosis................abortions in pregnant women.)

Line No. 95-96 need to rephrase

Line No. 121-122: how did you calculate total sample size 360? Why number of samples from sheep (n = 239) and in goats only (n = 121)

Line 128-134: add a reference

Line 141-142: Samples with an S/P% less than less than40% were considered negative; (less than in duplicate)

Line 135-172: add suitable references

Line 174-185: Results were not explained statistically.

In Tables 3,4,5, and 6, the authors analysed data through the Chi-square test, but I did not see a single Chi-square value

Line 249-250: Remove years Like Magalhães et al. (2016) [36] should be Magalhães et al. [36]. Same in lines No. 253-254

Line 283. Add the name of other species or rephrase the sentence

Author Response

General author response

We are greatly appreciating the constructive comments and suggestions from editor and all reviewers that greatly improved the quality of our manuscript. In this revision note, we have answered to comments through point by point revision. We have indicated author response in this revision note under title of Author’s response. Also, we indicated our changes and revisions in revision note and manuscript as blue-colored fonts. In addition, as a response to reviewer comment, we have created another file with newly added or deleted or modified information as tracking changes option. In addition, extensive English version and editing was carried out for the current version of manuscript. Hopefully, our revision and answers in below would be sufficient for publication of our study in your prestigious MDPI animals journal.

Reviewer #1

Comments and Suggestions for Authors

- Line No. 75-77 need to rephrase

Author’s response

We are greatly appreciating the constructive comments and suggestions from reviewer #1 that greatly improved the quality of our manuscript.

Rephrased as follows;

“A placental and / or foetal infection caused by T. gondii, N. caninum, or B. melitensis in pregnant sheep and goats may result in foetal death, abortion, or stillbirth, as well as severe placentitis in the case of brucellosis [12-17].” (lines 75-77)

- Double-check the reference given in line No. 79 (Reference No. 18)

Author’s response

More relevant reference was added instead of old one.

(18) Wang, X.; Lin, P.; Li, Y.; Xiang, C.; Yin, Y.; Chen, Z.; Du, Y.; Zhou, D.; Jin, Y.; Wang, A. Brucella suis vaccine strain 2 induces endoplasmic reticulum stress that affects intracellular replication in goat trophoblast cells in vitro. Front. Cell. Infect. Microbiol. 2016, 6, 19. doi: 10.3389/fcimb.2016.00019. (lines 377-379)

- Lines 86-92 are not relevant to the topic. Should be deleted (In humans, toxoplasmosis................abortions in pregnant women.)

Author’s response

This phrase has been deleted and clinical signs in human are referred briefly in the following sentence “Brucellosis mainly is a serious occupational hazard (characterized by variable clinical signs ranging from fever to miscarriage [23-25])” (lines 85-86)

- Line No. 95-96 need to rephrase

Author’s response

Rephrased as follows;

“Numerous animal species, including sheep and goats, are susceptible to toxoplasmosis, neosporosis, and brucellosis in Egypt [26-30]” (lines 89-90)

- Line No. 121-122: how did you calculate total sample size 360? Why number of samples from sheep (n = 239) and in goats only (n = 121)

Author’s response

The availability of samples with chosen criteria (seroprevalence of abortifacient agents among samples with and without history of abortion from sheep and goat) has determined the numbers and distribution of the used samples.

We added this information in the manuscript as follows:

“Selection of samples for the study was designed as to cover a large time span (5 years) and a wide geographical range (approximately 1000 kilometers apart) in order to give a reliable estimation of the epidemiological situation of the three pathogens. Furthermore, flocks with and without a history of abortion were included, while vaccination for Brucella was an exclusion criterium. Within the selected flocks, samples were collected randomly from apparently healthy animals.” (lines 118-123) 

- Line 128-134: add a reference

Author’s response

Reference no. 30 was added (Selim et al., 2021) (line 134)

- Line 141-142: Samples with an S/P% less than less than40% were considered negative; (less than in duplicate)

Author’s response

Corrected  (line143)

- Line 135-172: add suitable references

Author’s response

For testing the three pathogens, we have followed strictly the manufacturer’s instructions in which a detailed description was available including protocols, reagent preparation, measurement, calculation and interpretation procedures. (lines 136-163). 

- Line 174-185: Results were not explained statistically.

In Tables 3,4,5, and 6, the authors analysed data through the Chi-square test, but I did not see a single Chi-square value

Author’s response

We apologize for reviewer #1 for this confusing interpretation. This paragraph is purely descriptive for results of table 3, and we already added the prevalence rates either in numbers or in percentages and usually these are the most significant values in such kind of data. (lines 176-187)

In addition, in tables (3, 4, 5, 6) we added odd ratio and P value using chi square test via Graph prism software and vassarstat.net website. In such kind of epidemiological studies and when univariable analysis was conducted odd ratio and P value are the mainly used parameters for estimation of the gap or the difference among positive and negative results, as can be noticed in previous studies of our and other groups.

Fereig RM, Mohamed SGA, Mahmoud HYAH, AbouLaila MR, Guswanto A, Nguyen TT, Ahmed Mohamed AE, Inoue N, Igarashi I, Nishikawa Y. Seroprevalence of Babesia bovis, B. bigemina, Trypanosoma evansi, and Anaplasma marginale antibodies in cattle in southern Egypt. Ticks Tick Borne Dis. 2017 Jan;8(1):125-131. doi: 10.1016/j.ttbdis.2016.10.008. Epub 2016 Oct 20. PMID: 27789159.

Fereig RM, Abdelbaky HH, El-Alfy E-S,El-Diasty M, Elsayed A,Mahmoud HYAH, Ali AO, Ahmed A,Mossaad E, Alsayeqh AF and Frey CF(2022) Seroprevalence of Toxoplasmagondii and Neospora caninum incamels recently imported to Egyptfrom Sudan and a globalsystematic review.Front. Cell. Infect. Microbiol.12:1042279.doi: 10.3389/fcimb.2022.1042279

Fereig RM, Abdelbaky HH, Mazeed AM, El-Alfy ES, Saleh S, Omar MA, Alsayeqh AF, Frey CF. Prevalence of Neospora caninum and Toxoplasma gondii Antibodies and DNA in Raw Milk of Various Ruminants in Egypt. Pathogens. 2022 Nov 7;11(11):1305. doi: 10.3390/pathogens11111305. PMID: 36365056.

Udonsom R, Nishikawa Y, Fereig RM, Topisit T, Kulkaweewut N, Chanamrung S, Jirapattharasate C. Exposure to Toxoplasma gondii in Asian Elephants (Elephas maximus indicus) in Thailand. Pathogens. 2021 Dec 21;11(1):2. doi: 10.3390/pathogens11010002. PMID: 35055950; PMCID: PMC8778166.

Pagmadulam B, Myagmarsuren P, Fereig RM, Igarashi M, Yokoyama N, Battsetseg B, Nishikawa Y. Seroprevalence of Toxoplasma gondii and Neospora caninum infections in cattle in Mongolia. Vet Parasitol Reg Stud Reports. 2018 Dec;14:11-17. doi: 10.1016/j.vprsr.2018.08.001. Epub 2018 Aug 18. PMID: 31014714.

However, we also can do any other kind of statistical analyses based on the reviewer 1 specification.

- Line 249-250: Remove years Like Magalhães et al. (2016) [36] should be Magalhães et al. [36]. Same in lines No. 253-254

Author’s response

Years have been deleted (lines 251-255)

- Line 283. Add the name of other species or rephrase the sentence

Author’s response

Revised as follows;

“This area is endemic for different Brucella species including B. abortus, the causative agent of bovine brucellosis, and B. melitensis, the main cause of ovine and caprine brucellosis [26,29].” (lines 286-288)

Reviewer 2 Report

In this study, authors have investigated the seroprevalence of T. gondii and N. caninum, and Brucella spp in sheep and goats in four regions in Egypt between 2016 to 2021. A total of 360 blood sera was screened using ELISA to detect specific antibodies against texted pathogens. Meanwhile, a questionnaire survey was performed to collect information on potential risk factors of the infection in each pathogenes.

The study reported Seropositivity rates of 46.1%, 11.9%, 35 and 8.6% for T. gondii, N. caninum, and Brucella spp., respectively. In addition, the animal with a history of abortion seems to be a risk factor for brucella infection, while not considered a risk factor in other pathogens.   The data presented shows no doubt about the seroprevalence of T. gondii and N. caninum and brucella spp in sheep and goats and their associated risk factors. However, some issues should be addressed before considering this manuscript for publication. Specific comments are below:

1.       Line 118: samples were collected between 2016 and 2021. Why was this variation on time? Any specific reason?

2.       Line 121: How did the author decide on sample size 360? Based on what and what is methods used to collect this sample? Please explain more

3.       Line 133: Please give detailed information on the process of serum centrifugation speed and time.

4.       The study limitation should be highlighted

Author Response

Reviewer #2

Comments and Suggestions for Authors

In this study, authors have investigated the seroprevalence of T. gondii and N. caninum, and Brucella spp in sheep and goats in four regions in Egypt between 2016 to 2021. A total of 360 blood sera was screened using ELISA to detect specific antibodies against texted pathogens. Meanwhile, a questionnaire survey was performed to collect information on potential risk factors of the infection in each pathogenes.

The study reported Seropositivity rates of 46.1%, 11.9%, 35 and 8.6% for T. gondiiN. caninum, and Brucella spp., respectively. In addition, the animal with a history of abortion seems to be a risk factor for brucella infection, while not considered a risk factor in other pathogens.   The data presented shows no doubt about the seroprevalence of T. gondii and N. caninum and brucella spp in sheep and goats and their associated risk factors. However, some issues should be addressed before considering this manuscript for publication. Specific comments are below:

  1. Line 118: samples were collected between 2016 and 2021. Why was this variation on time? Any specific reason?

Author’s response

We are greatly appreciating the constructive comments and suggestions from reviewer #2 that greatly improved the quality of our manuscript.

“The availability of samples with chosen criteria (seroprevalence of abortifacient agents among samples with and without history of abortion from sheep and goat) has determined the numbers and distribution of the used samples.

We added this information in the manuscript as follows:

“Selection of samples for the study was designed as to cover a large time span (5 years) and a wide geographical range (approximately 1000 kilometers apart) in order to give a reliable estimation of the epidemiological situation of the three pathogens. Furthermore, flocks with and without a history of abortion were included, while vaccination for Brucella was an exclusion criterium. Within the selected flocks, samples were collected randomly from apparently healthy animals.” (lines 118-123) 

  1. Line 121: How did the author decide on sample size 360? Based on what and what is methods used to collect this sample? Please explain more

Author’s response

Same as mentioned in earlier response, the availability of samples with chosen criteria (seroprevalence of abortifacient agents among samples with and without history of abortion from sheep and goat) has determined the numbers and distribution of the used samples.

We added this information in the manuscript as follows:

“Selection of samples for the study was designed as to cover a large time span (5 years) and a wide geographical range (approximately 1000 kilometers apart) in order to give a reliable estimation of the epidemiological situation of the three pathogens. Furthermore, flocks with and without a history of abortion were included, while vaccination for Brucella was an exclusion criterium. Within the selected flocks, samples were collected randomly from apparently healthy animals.” (lines 118-123) 

  1. Line 133: Please give detailed information on the process of serum centrifugation speed and time.

Author’s response

Centrifugation details were added “centrifuged at 10.000xg for 15 min…..ref 30 (lines 134)

  1. The study limitation should be highlighted

Author’s response

Mentioned as follows;

“Due to the limitations in the number, content, and distribution of studies focusing on the relevant theme worldwide including Egypt, we could not perform a comprehensive comparison with our study. Additionally, higher numbers of samples are required from sheep, goats and other species and geographic locations to establish an accurate assessment for the main causes of abortions among our tested pathogens.” (lines 302-306)

Round 2

Reviewer 1 Report

Accepted in the present form.